# Using the Edmonton Symptom Assessment System (ESAS) to Describe Symptom Burden Associated with Breast Cancer and Related Treatments: A Cross-Sectional Study

**DOI:** 10.3390/curroncol32110598

**Published:** 2025-10-24

**Authors:** Sofia Torres, Maureen Trudeau, Geoffrey Liu, Nicholas Mitsakakis, Ahmed M. Bayoumi

**Affiliations:** 1Institute of Health Policy, Management and Evaluation, University of Toronto, Toronto, ON M5T 3M6, Canada; ahmed.bayoumi@utoronto.ca; 2Division of Medical Oncology and Malignant Hematology, Odette Cancer Center, Sunnybrook Health Sciences Centre, Toronto, ON M4N 3M5, Canada; maureen.trudeau@sunnybrook.ca; 3Sunnybrook Research Institute, Toronto, ON M4N 3M5, Canada; 4Division of Medical Oncology and Hematology, Department of Medicine, Princess Margaret Cancer Centre/University Health Network, University of Toronto, Toronto, ON M5G 2C4, Canada; geoffrey.liu@uhn.ca; 5Children’s Hospital of Eastern Ontario Research Institute, Ottawa, ON K1H 8L1, Canada; n.mitsakakis@theta.utoronto.ca; 6Division of Biostatistics, Dalla Lana School of Public Health, University of Toronto, Toronto, ON M5S 3E3, Canada; 7MAP Centre for Urban Health Solutions, Li Ka Shing Knowledge Institute, St. Michael’s Hospital, Toronto, ON M5B 1X3, Canada; 8Division of General Internal Medicine, St. Michael’s Hospital, Toronto, ON M5B 1W8, Canada; 9Department of Medicine, University of Toronto, Toronto, ON M5S 3H2, Canada; 10Institute of Medical Sciences, University of Toronto, Toronto, ON M5S 3K3, Canada

**Keywords:** breast cancer, Edmonton Symptom Assessment System (ESAS), patient reported outcomes

## Abstract

Breast cancer is the most common cancer among women. Many patients experience symptoms that affect their ability to function, including feeling tired, having low energy, experiencing anxiety, and an overall sense of not feeling well. While some women report only mild discomfort, many have symptoms that are moderate to severe and may require medical attention. In our study, we asked women who had a range of stages of breast cancer to report their symptoms and daily functioning. We found that about 7 out of 10 had at least one symptom that was strong enough to need extra care. Importantly, symptoms were common for both women with advanced breast cancer and for those with earlier-stage disease, who were expected to be cured. These findings show how important it is for doctors and nurses to ask patients directly about their symptoms during clinic visits. Regular symptom checks can help healthcare teams identify who needs more support, provide timely treatment, and improve quality of life for women living with breast cancer.

## 1. Introduction

Breast cancer is the most frequently diagnosed malignancy among women across the world and, in high-income countries, the second leading cause of cancer-related mortality [1]. Breast cancer survival has improved significantly in recent decades, as a result of screening programmes and therapeutic strategies. For example, in Canada, the five-year net survival for breast cancer in women is 89%, ranging from nearly 100% for women with stage I disease to 23% in women with stage IV disease [2]. With increasing survival has come the need for a better understanding of the frequency and impact of symptoms that women experience– due to both disease sequelae and treatment [3,4,5,6]. These symptoms, which may be common, can fluctuate over time and substantially impair functional capacity [7]. For example, depression has been reported in up to 66% of breast cancer survivors, with risk peaking one year after diagnosis [8]. Symptom clusters, particularly those encompassing emotional distress, can adversely affect functional status and quality of life [9,10,11]. Accurate identification of symptom burden is necessary for the development of tailored management strategies.

Patient-reported outcomes (PROs) are direct accounts from patients about their health status; patient-reported outcome measures (PROMs) are validated instruments designed for the systematic collection of such data [12]. The Edmonton Symptom Assessment System (ESAS) is a PROM that measures common cancer-related symptoms including pain, fatigue, nausea, depression, and anxiety [13,14]. Patients with high physical and total symptom scores, as assessed by ESAS, have shorter survival [15]. In Ontario, routine collection of ESAS at cancer centres was initiated in 2007. Results are immediately available to clinical teams for potential discussion during outpatient visits. Another PROM, the Patient-Reported Functional Status tool, has also been incorporated into routine assessments since 2013 [16]. Approximately 60% of eligible breast cancer patients routinely complete these symptom questionnaires during clinic visits [17].

Routine use of PROMs has been associated with improved symptom identification and monitoring, enhanced communication between patients and their care team, better quality of life, and, in some studies, improved survival [18,19,20,21,22,23,24]. Nevertheless, few studies have reported ESAS outcomes in breast cancer populations [25,26]. To our knowledge, only one previous study has used ESAS to characterize the symptom burden of breast cancer patients across all stages and with a range of times since diagnosis [27].

As part of a broader investigation into quality of life among women with a history of breast cancer, we aimed to both describe and interpret ESAS scores across the disease spectrum. Our interpretation is based on applying clinically significant thresholds to ESAS scores, specifically, levels that may warrant clinical intervention and are therefore a clinically meaningful assessment of patients’ needs. In addition, we measured patient-reported functional status to characterize functional impact in parallel with symptom burden. To our knowledge, this is the first study to use this approach across the full disease spectrum, offering novel insights to guide supportive care strategies and inform future clinical practice.

## 2. Materials and Methods

### 2.1. Research Design and Procedures

We conducted a cross-sectional study with a convenience sample of women with invasive breast cancer who were recruited during outpatient visits between November 2016 and June 2017. Participants completed ESAS and the Patient-Reported Functional Status tool, either in waiting room kiosks or on electronic tablets (iPads^®^, Apple Inc., Cupertino, CA, USA). Clinical and demographic data were obtained from patient self-report and supplemented by chart review. The study was reported in accordance with the STROBE guidelines (Appendix A) [28].

### 2.2. Setting and Participants

The study was conducted at two tertiary cancer centres in Toronto, Canada: the Louise Termerty Breast Cancer Centre and the Princess Margaret Cancer Centre. Eligible participants were women aged ≥18 years with a histologically confirmed diagnosis of invasive breast cancer (stage I to IV), who were undergoing treatment or active surveillance at the time of recruitment and were able to read and complete the questionnaires in English. Women were excluded if they could not provide informed consent or complete the questionnaires independently. Each participant was enrolled only once. Patients were recruited consecutively during the study period.

### 2.3. Symptom and Functional Status Assessment

ESAS is a validated self-reported instrument that measures nine common cancer-related symptoms: pain, tiredness, drowsiness, nausea, lack of appetite, shortness of breath, depression, anxiety, and overall well-being [13,14]. It requires approximately 3 min to complete [17]. Each symptom is scored from 0 (no symptoms) to 10 (worst possible symptom). For clinical interpretation, scores can be categorized as none (0), mild (1–3), moderate (4–6) and severe (7–10) [29,30,31]. Moderate to severe scores (≥4) are considered clinically significant, generally warranting intervention such as a prescription, additional investigation, or referral to a specialist [29,31,32].

In addition to symptoms scores, we calculated three ESAS summary scores: (i) a physical distress score (0 to 60; sum of pain, tiredness, nausea, drowsiness, lack of appetite, and shortness of breath), (ii) an emotional distress score (0 to 20; sum of anxiety and depression), and (iii) a total symptom distress score (0–90; sum of physical and emotional scores plus overall well-being) [30,33,34,35].

Functional status was assessed using the Patient-Reported Functional Status tool, a validated patient-completed version of the Eastern Cooperative Oncology Group (ECOG) performance status instrument. [16] This tool is a single-item measure with which patients rate their activity over the past month on a five-point scale ranging from 0 (“normal with no limitation”) to 4 (“pretty much bedridden, rarely out of bed”). The ECOG performance status is widely used by oncologists to quantify functional capacity and to estimate prognosis [36].

### 2.4. Sample Size

We did not calculate sample size a priori because our study was a secondary analysis of a fixed convenience sample. We included all consecutive eligible patients approached during the study period.

### 2.5. Statistical Analysis

We used descriptive statistics to characterize the study population as a whole and stratified by ESAS symptom severity and Patient-Reported Functional Status tool level. Comparisons between patients who completed ESAS and patients who did not were made using chi-squared or Fisher’s exact test, as appropriate.

Multivariable logistic regression was used to identify socio-demographic and clinical characteristics independently associated with having at least one clinically significant symptom (ESAS score ≥4). Candidate predictors were selected a priori based on clinical relevance, including age, Charlson Comorbidity Index, breast cancer subtype, disease status (metastatic vs. non-metastatic), and treatment type.

All statistical analyses were conducted using SAS^®^ 9.4 (SAS Institute, Cary, NC, USA). All tests were two-sided, with *p* < 0.05 considered statistically significant.

### 2.6. Ethical Considerations

The study was conducted in accordance with the Declaration of Helsinki. Ethical approval was obtained from the Sunnybrook Research Ethics Board (protocol number: 2551; approval date: 28 October 2016) and the University Health Network Research Ethics Boards (protocol number: 06-639: 6 December 2006). Written informed consent was obtained from all participants prior to participation. Confidentiality was ensured through secure storage, restricted access, and data de-identification before analysis.

## 3. Results

Of 549 women recruited for the larger study, 381 (69%) completed ESAS during their clinic visit (Table 1). Respondents had a mean age of 56.8 years (SD 12); most had no comorbidities (Charlson Comorbidity Index of 0), had a post-secondary education, and were born outside Canada. Approximately three-quarters were receiving active treatment, most commonly hormonal therapy. Compared with respondents, women who did not complete ESAS were more likely to be pre-menopausal, never married, and White, and had a different distribution of diagnosis years.

Overall, 70% of patients reported at least one moderate to severe symptom, and 19% reported at least one severe symptom (Table 2). Tiredness, lack of well-being, anxiety, and depression were the most frequently reported symptoms, while nausea was the least common (2–8% across groups; Figure 1). Lack of well-being was the most common symptom among patients with metastatic breast cancer (44%), whereas tiredness was most common among women without metastatic disease (25 to 32%).

Mean ESAS physical, emotional, and total Symptom Distress Scores (SDS) were low across all disease groups, including patients with metastatic disease (Table 3). Despite this, 81% of patients with metastatic breast cancer reported at least one moderate to severe symptom.

Most patients (86%) reported an ECOG performance status of 0 (“normal with no limitation”) or 1 (“not my normal self, but able to be up and about with fairly normal activities”) (Table 4). Nearly half of patients with functional limitations (ECOG ≥ 2) had metastatic disease.

In multivariable analysis, being in the first year after primary breast cancer was associated with a lower probability of reporting at least one clinically significant symptom compared with metastatic disease (Odds ratio [OR] 0.49, 95% CI 0.24–0.90, Table 5). Patients diagnosed ≥6 years earlier also had a reduced likelihood, although this result was less precise (OR 0.50, 95% CI 0.22–1.13).

## 4. Discussion

We evaluated symptom burden and functional status among women with breast cancer using ESAS and the Patient-Reported Functional Status tool. By focusing on clinically significant thresholds rather than mean scores, we aimed to provide a more meaningful assessment of symptoms that require intervention. Our cohort included both patients with potentially curable disease and those with metastatic breast cancer, thus capturing the full spectrum of the disease continuum.

We found that 70% of patients reported at least one moderate to severe symptom, and nearly one in five reported at least one severe symptom. Fatigue, lack of well-being, anxiety, and depression were the most frequently reported symptoms, while nausea was least common. Importantly, two-thirds of patients with curable disease also reported at least one clinically significant symptom, indicating that symptom burden is not confined to those with advanced disease. Even women who were disease-free more than six years after diagnosis frequently reported persistent symptoms, particularly fatigue, lack of well-being, anxiety, and depression.

Our findings are consistent with systematic reviews reporting that fatigue, emotional distress, and depressive symptoms are among the most common and persistent issues faced by breast cancer survivors, sometimes lasting for years after treatment completion [10,11,37]. Prior studies using ESAS in breast cancer populations have identified similar symptom patterns, despite recruiting diverse patient groups with a range of disease durations [27,38]. By applying clinically significant thresholds, our analysis demonstrates how focusing on group-level averages may underestimate the true prevalence of symptoms requiring clinical attention.

ESAS provides a brief, practical tool to screen symptoms such anxiety and depression during oncology visits, and to identify patients in need of further evaluation and support [39,40]. Interventions such as structured exercise programmes have been shown to reduce fatigue, anxiety, and depressive symptoms, in breast cancer survivors and are recommended by clinical guidelines [41,42,43]. Moreover, as patients transition from cancer centres to community-based care, individualized survivorship care plans may facilitate ongoing symptom management and improve long-term outcomes [44].

Effective symptom management requires a multidisciplinary team [45,46]. Oncologists, nurses, psychologists, social workers, physiotherapists, and palliative care specialists all contribute to the identification and treatment of symptom burden. Integrating PROMs such as ESAS into multidisciplinary care pathways can enhance communication, guide individualized interventions, and ensure timely referral to appropriate services.

Our study has several strengths. We drew on a relatively large cohort of consecutively recruited patients from two major cancer centres, which enhances generalizability. We used validated PROMs that are routinely integrated into clinical practice, and we applied clinically relevant thresholds to provide a more accurate assessment of symptoms requiring intervention. The inclusion of ECOG performance status provided additional insight into the relationship between symptom burden and functional capacity.

Limitations include the cross-sectional design, which precludes causal inference, and the exclusion of non-English speakers, which may limit generalizability. ESAS does not capture disease- or treatment-specific symptoms, such as cognitive impairment, lymphedema, or sexual dysfunction, which are nonetheless common and clinically relevant [10,47]. In men, the recognition that some patients reported symptoms not addressed by ESAS led to the implementation of the Expanded Prostate Cancer Index Composite for clinical practice (EPIC-CP) in Ontario [48,49]. A similar approach may be beneficial for addressing the range of symptoms experienced by women with a history of breast cancer. Selection bias is possible, as non-responders differed from responders in some characteristics, including race and ethnicity. The vast majority of ESAS studies includes mostly White patients, and mean scores may vary with ethno-racial background [50]. Moreover, patients too unwell to attend outpatient clinics (for example, patients admitted to hospital or to palliative care) or long-term survivors no longer followed in cancer centres were not included. Finally, the sample size may have been too small to estimate some important effects with precision.

Our results have important implications for clinical practice. Routine use of PROMs with attention to clinically significant thresholds can help clinicians identify patients who are most in need of intervention. Incorporating ESAS into survivorship care planning and community-based follow-up may improve continuity of care. Importantly, multidisciplinary involvement can address the complex and overlapping physical and psychological symptoms experienced by breast cancer patients.

Future research should investigate longitudinal symptom trajectories, evaluate interventions tailored to clinically significant symptom thresholds, and explore differences in symptom burden across ethno-racial groups. Research should also assess the impact of PROM-driven multidisciplinary care on long-term outcomes such as quality of life, healthcare utilization, and survival.

## 5. Conclusions

In conclusion, 70% of women with breast cancer reported at least one moderate to severe symptom, and nearly 20% reported at least one severe symptom, as measured by ESAS. Fatigue, lack of well-being, anxiety, and depression were the most common symptoms, and importantly, a substantial proportion of patients with potentially curable disease also reported clinically significant burden. These findings highlight that symptom management remains an important need across the entire disease spectrum, not only for patients with metastatic disease.

Our results have important implications for clinical practice. Routine use of PROMs such as ESAS, with attention to clinically significant thresholds, may improve the timely identification of patients in need of intervention. Incorporating symptom assessment into survivorship care planning and multidisciplinary care pathways can enhance quality of life and ensure continuity of care.

Future research should focus on whether symptom-specific interventions are effective and are associated with better functional outcomes and overall quality of life in breast cancer survivors.

## Figures and Tables

**Figure 1 curroncol-32-00598-f001:**
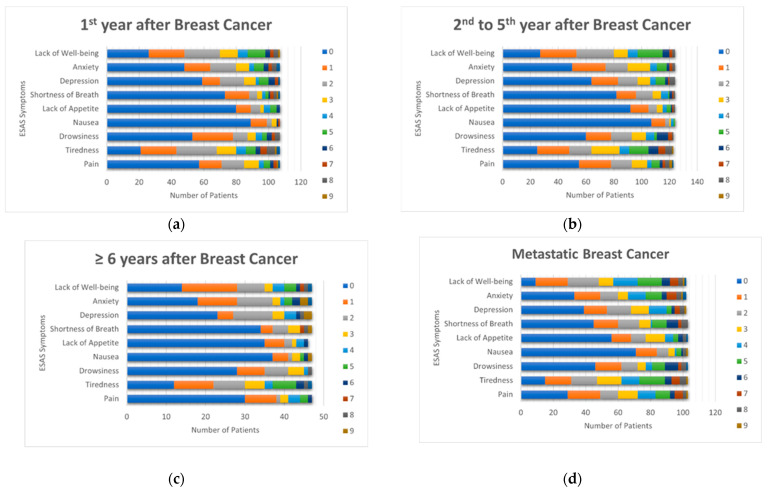
ESAS Symptom Scores by disease status. (**a**) First year after breast cancer diagnosis; (**b**) Second to fifth year after breast cancer diagnosis; (**c**) Sixth and following years after breast cancer diagnosis; (**d**) Metastatic breast cancer.

**Table 1 curroncol-32-00598-t001:** Characteristics of patients who completed ESAS.

Characteristics	Answered ESAS *N* = 381 (%)	Did Not Answer ESAS *N* = 168 (%)	*p*-Value
Age (Mean; SD; min-max)	56.8 (12; 25–87)	56.5 (13; 29–90)	0.802
Age <45 years 45–64 years ≥65 years	55 (14%) 223 (59%) 103 (27%)	33 (20%) 90 (54%) 45 (27%)	0.290
Menopausal Status Pre-menopausal Post-menopausal Unsure/Answer missing	88 (23%) 257 (67%) 36 (9%)	58 (35%) 100 (60%) 10 (6%)	0.014
Charlson Comorbidity Index 0 1–2 ≥3	274 (72%) 92 (24%) 15 (4%)	122 (73%) 40 (24%) 6 (4%)	0.973
Born in Canada Yes No Preferred not to answer	155 (41%) 219 (57%) 7 (2%)	83 (49%) 81 (48%) 4 (2%)	0.132
Annual Family Income ^1^ $0 to $59,999 ≥ $60,000 Does not know/Prefers not to answer	102 (27%) 179 (47%) 100 (26%)	40 (24%) 86 (51%) 42 (25%)	0.640
Education Below Grade 8 Attended/graduated high school Attended/graduated college/university Postgraduate/professional Missing	12 (3%) 66 (17%) 215 (56%) 84 (22%) 4 (1%)	3 (2%) 18 (11%) 98 (58%) 45 (27%) 4 (2%)	0.150
Employment Status Retired Unemployed Employed Other (e.g., on leave, disability) Missing	118 (31%) 38 (10%) 164 (43%) 60 (16%) 1 (0.3%)	48 (29%) 14 (8%) 78 (46%) 26 (15%) 2 (1%)	0.609
Marital Status Married/common law Separated/divorced/widowed Single/Never married Missing	261 (69%) 81 (21%) 36 (9%) 3 (1%)	109 (65%) 27 (16%) 29 (17%) 3 (2%)	0.031
Primary Language spoken at home English French Other Missing	273 (72%) 4 (1%) 103 (27%) 1 (0.3%)	120 (71%) 3 (2%) 42 (25%) 3 (2%)	0.222
Racial or ethnic group Asian-East/South East Asian-South Black/African Canadian Caribbean Latin American Middle Eastern Mixed heritage White First Nations Preferred Not to Answer	77 (20%) 22 (6%) 2 (1%) 19 (5%) 10 (3%) 29 (8%) 3 (1%) 210 (55%) 0 (0%) 9 (2%)	32 (19%) 9 (5%) 5 (3%) 3 (2%) 1 (1%) 3 (2%) 0 (0%) 111 (66%) 2 (1%) 2 (1%)	0.002
Mean number of years living with invasive breast cancer (95% CI)	4.7 years (4.1–5.3)	5.0 years (4.0–6.0)	0.633
Year of Primary Diagnosis 1985–2000 2001–2005 2006–2010 2011–2015 2016–2017	22 (6%) 23 (6%) 55 (14%) 152 (40%) 129 (34%)	16 (10%) 12 (7%) 9 (5%) 68 (40%) 63 (38%)	0.024
Stage at diagnosis Stage I Stage II Stage III Stage IV Unknown	122 (32%) 159 (42%) 71 (19%) 26 (7%) 3 (1%)	52 (31%) 67 (40%) 32 (19%) 14 (8%) 3 (2%)	0.918
Breast Cancer Subtype Hormone-Receptor Positive HER2 Positive Triple negative	226 (60%) 74 (20%) 76 (20%)	106 (63%) 35 (21%) 27 (16%)	0.522
Previous Breast Surgery	343 (90%)	146 (87%)	0.280
Current radiotherapy	6 (2%)	6 (4%)	0.140
Current systemic therapy Chemotherapy (+/− targeted therapy) Hormonal treatment (+/− targeted therapy) Targeted therapy (only)	63 (17%) 204 (54%) 17 (4%)	23 (14%) 87 (52%) 7 (4%)	0.679
Disease Status 1st year after primary breast cancer 2nd to 5th year after primary breast cancer 6th and following years after primary breast cancer Metastatic Breast Cancer	107 (28%) 124 (33%) 47 (12%) 103 (27%)	52 (31%) 61 (36%) 15 (9%) 40 (24%)	0.478

The study sites asked about income using different categories. These categories were collapsed and combined as shown. Abbreviations: 95% CI = 95% confidence interval; max = maximum; min = minimum; N = number; % = percentage; SD = standard deviation.

**Table 2 curroncol-32-00598-t002:** Frequency of moderate to severe ESAS symptom scores (≥4).

ESAS Symptoms (Moderate to Severe Scores, 4–10)	1st Year After Primary Breast Cancer (*N* = 107)	2nd to 5th Year After Primary Breast Cancer (*N* = 124)	≥6 Years After Primary Breast Cancer (*N* = 47)	Metastatic Breast Cancer (*N* = 103)	Total (*N* = 381)
Pain	13 (12%)	19 (15%)	6 (13%)	31 (30%)	69 (18%)
Tiredness	27 (25%)	39 (32%)	12 (26%)	41 (40%)	119 (31%)
Drowsiness	15 (14%)	20 (16%)	2 (4%)	26 (25%)	63 (17%)
Nausea	2 (2%)	3 (2%)	3 (6%)	8 (8%)	16 (4%)
Lack of Appetite	10 (9%)	9 (7%)	3 (6%)	14 (14%)	36 (9%)
Shortness of Breath	11 (10%)	10 (8%)	3 (6%)	23 (22%)	47 (12%)
Depression	15 (14%)	18 (15%)	7 (15%)	23 (23%)	63 (17%)
Anxiety	19 (18%)	18 (15%)	8 (17%)	36 (35%)	81 (21%)
Lack of Well-being	26 (24%)	34 (27%)	10 (21%)	45 (44%)	115 (30%)
Patients who reported at least 1 moderate or severe symptom (4–10)	66 (62%)	88 (71%)	30 (64%)	83 (81%)	267 (70%)
Patients who reported at least 1 severe symptom (7–10)	20 (19%)	21 (17%)	6 (13%)	26 (25%)	73 (19%)

Abbreviations: N = number; % = percentage.

**Table 3 curroncol-32-00598-t003:** ESAS summary scores by patient group.

ESAS Scores (Mean, SD, Min–Max)	1st Year After Primary Breast Cancer (*N* = 107)	2nd to 5th Year After Primary Breast Cancer (*N* = 124)	≥6 Years After Primary Breast Cancer (*N* = 47)	Metastatic Breast Cancer (*N* = 103)	Total (*N* = 381)
ESAS Physical Score	7.2 (8.3; 0–40)	7.5 (7.7; 0–35)	6.3 (8.0; 0–43)	11.5 (10.3; 0–51)	8.3 (8.9; 0–51)
ESAS Emotional Score	3.1 (4.2; 0–20)	2.9 (3.6; 0–14)	3.5 (4.7; 0–18)	4.4 (4.5; 0–18)	3.5 (4.2; 0–20)
ESAS Total SDS	12.6 (12.6; 0–64)	12.7 11.8; 0–49)	11.8 (13.8; 0–66)	19.1 (15.1; 0–78)	14.3 (13.5; 0–78)

Abbreviations: Max = maximum; Min = minimum; N = number; SD = standard deviation.

**Table 4 curroncol-32-00598-t004:** Patient-Reported Functional Status (ECOG) results.

Patient-Reported Functional Status Scores	1st Year After Primary Breast Cancer (*N* = 106)	2nd to 5th Year After Primary Breast Cancer (*N* = 123)	≥6 Years After primary Breast Cancer (*N* = 46)	Metastatic Breast Cancer (*N* = 103)	Total (*N* = 378)
Normal with no limitations (0)	33 (31%)	56 (46%)	29 (63%)	29 (28%)	147 (39%)
Not my normal self, but able to be up and about with fairly normal activities (1)	56 (52%)	57 (46%)	15 (33%)	52 (50%)	180 (48%)
Not feeling up to most things, but in bed or chair less than half the day (2)	13 (12%)	7 (6%)	1 (2%)	17 (17%)	38 (10%)
Able to do little activity and spend most of the day in bed or chair (3)	4 (4%)	3 (2%)	1 (2%)	5 (5%)	13 (3%)
Pretty much bedridden, rarely out of bed (4)	-	-	-	-	-

Abbreviations: N = number; % = percentage.

**Table 5 curroncol-32-00598-t005:** Logistic regression model for predictors of at least one clinically significant ESAS symptom.

Predictor	*N* = 381 (%)	Odds Ratio (95% CI)	*p*-Value
**Age**			
<45 years	55 (14%)	0.86 (0.40, 1.84)	0.694
45–64 years	223 (59%)	0.10 (0.58, 1.73)	0.991
≥65 years	103 (27%)	Reference	-
**Charlson Comorbidity Index**			
0	274 (72%)	0.95 (0.28, 3.26)	0.933
1–2	92 (24%)	1.26 (0.35, 4.55)	0.728
≥3	15 (4%)	Reference	-
**Breast Cancer Subtype**			
Hormone-Receptor Positive	226 (60%)	1.26 (0.70, 2.27)	0.439
Triple negative	76 (20%)	Reference	-
HER2 Positive	74 (20%)	1.21 (0.59, 2.49)	0.606
**Breast Cancer Status**			
1st year after primary BC	107 (28%)	0.49 (0.24, 0.90)	0.023
2nd to 5th year after primary BC	124 (33%)	0.63 (0.32, 1.21)	0.161
6th and following years after primary BC	47 (12%)	0.50 (0.22, 1.13)	0.094
Metastatic BC	103 (27%)	Reference	-
**Breast Cancer Surgery**			
Yes	243 (90%)	Reference	-
No	38 (10%)	0.95 (0.43, 2.10)	0.901
**Current systemic therapy**			
Yes	283 (74%)	Reference	-
No	98 (26%)	0.64 (0.38, 1.08)	0.095

Abbreviations: BC = Breast Cancer; 95% CI = 95% confidence interval; N = number; % = percentage.

## Data Availability

All data generated or analyzed during this study are included in this article.

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
