# Peer review of "Using the Edmonton Symptom Assessment System (ESAS) to Describe Symptom Burden Associated with Breast Cancer and Related Treatments: A Cross-Sectional Study"

_curroncol, 2025, doi:10.3390/curroncol32110598_

Round 1
Reviewer 1 Report
Comments and Suggestions for Authors
This study provides an interesting description of the symptom burden trajectories in patients with breast cancer. Please find below my specific comments:
Title
Please remove “diagnosis” after breast cancer since it appears redundant.
Introduction
I suggest adding a paragraph (after the first) regarding symptom burden in breast cancer patients to highlight the phenomenon’s relevance.
I suggest starting the second paragraph by describing the PROs and then focusing on ESAS. This approach will help to contextualize these instruments.
Please provide the most recent references for epidemiological data.
Methods
Please clearly state the study’s design.
Please add a subsection with eligibility criteria and sampling method. In this regard, the authors should precisely clarify the cohort of patients they intended to consider.
The authors should report and reference the guidelines adopted for the study’s reporting. Then, be sure that the reporting is consistent with the mentioned guidelines.
Sample size calculation is recommended.
I suggest adding a subsection with ethical considerations, including how the consent form was obtained, the information provided to participants, and how data confidentiality was guaranteed.
Institutional Review Board Statement
Please provide the ethical approval number of the study along with the date.
Discussion
Overall, this section should be revised. The first paragraph should summarise the study’s rationale and aim. After that, the authors can report the study’s results and systematically discuss them with the available literature. The structure should follow a logical flow and should be consistent with the results’ structure. This approach will improve the manuscript’s readability.
Conslusions
This section should report the main findings. The authors generally reported that they identified symptoms, but the reported information does not inform the reader about the main findings.
I also suggest adding a paragraph with the study’s implications.
References
Please revise references and provide the most recent when feasible. Several references are too old.
The full stop should be placed after the references’ brackets. Please revise the entire text.
Author Response
Thank you very much for taking the time to review this manuscript. Please find the detailed responses below and the corresponding revisions in track changes in the re-submitted files.
Comment 1: “This study provides an interesting description of the symptom burden trajectories in patients with breast cancer.”
Response 1: We thank the reviewer for this positive comment.
Comment 2: “Title: Please remove “diagnosis” after breast cancer since it appears redundant.”
Response 2: We agree. In revising the title, we also aligned it with the STROBE guidelines, which recommend specifying the study design. The title now reads: “Using the Edmonton Symptom Assessment System (ESAS) to describe symptom burden associated with breast cancer and related treatments: a cross-sectional study.” This revision removes redundancy and improves methodological transparency.
Comment 3: “Introduction- I suggest adding a paragraph (after the first) regarding symptom burden in breast cancer patients to highlight the phenomenon’s relevance.”
Response 3: We agree and have added text after the first paragraph (lines 91–96) highlighting the relevance of symptom burden in breast cancer, including its impact on functional capacity, emotional health, and quality of life.
Comment 4: “Introduction- I suggest starting the second paragraph by describing the PROs and then focusing on ESAS. This approach will help to contextualize these instruments.”
Response 4: We agree. We revised the paragraph to introduce PROs and PROMs before describing ESAS (lines 97-99).
Comment 5: “Introduction- Please provide the most recent references for epidemiological data.”
Response 5: We agree and have updated the epidemiological references in first paragraph of the Introduction to include the most recent available data. References 1, 3 and 4 were updated.
Comment 6: “Methods- Please clearly state the study’s design.”
Response 6: We agree. Section 2.1 has been revised to explicitly state (lines 135 to 137): “We conducted a cross-sectional study with a convenience sample of women with invasive breast cancer who were recruited during outpatient visits between November 2016 and June 2017.”
Comment 7: “Methods- Please add a subsection with eligibility criteria and sampling method. In this regard, the authors should precisely clarify the cohort of patients they intended to consider.”
Response 7: We agree. Section 2.2 (Setting and Participants, line 145–153) now specifies the inclusion/exclusion criteria and explains that patients were recruited consecutively during the study period.
Comment 8: “Methods- The authors should report and reference the guidelines adopted for the study’s reporting. Then, be sure that the reporting is consistent with the mentioned guidelines.”
Response 8: We agree. Section 2.1 (Research design and procedures, lines 142-143) now references the STROBE guidelines. We ensured the manuscript is consistent with these standards (please see Table S1 submitted as supplementary information).
Comment 9: “Methods- Sample size calculation is recommended.”
Response 9: As this was a secondary analysis of a fixed convenience sample, a priori sample-size calculation was not applicable. This has been clarified in Section 2.4 (lines 188 to 190).
Comment 10: “Methods- I suggest adding a subsection with ethical considerations, including how the consent form was obtained, the information provided to participants, and how data confidentiality was guaranteed.”
Response 10: We agree. Section 2.6 (Ethical considerations, lines 209 to 215) now details ethics approval, the informed consent process, and measures taken to ensure data confidentiality.
Comment 11: “Institutional Review Board Statement: Please provide the ethical approval number of the study along with the date.”
Response 11: We have included this information in Section 2.6 (Ethical considerations, lines 209 to 215). The revised text states: “Ethical approval was obtained from the Sunnybrook Research Ethics Board (protocol 2551; approval date: 2016-10-28) and the University Health Network Research Ethics Board (protocol 06-639; approval date: 2006-12-06).”
Comment 12: “Discussion- Overall, this section should be revised. The first paragraph should summarise the study’s rationale and aim. After that, the authors can report the study’s results and systematically discuss them with the available literature. The structure should follow a logical flow and should be consistent with the results’ structure. This approach will improve the manuscript’s readability.”
Response 12: We agree. The Discussion section has been fully restructured. It now begins with the study’s rationale and aim, followed by key findings, comparisons with prior studies and systematic reviews, interpretation of clinical relevance, the role of multidisciplinary care, strengths and limitations, and implications for practice and research.
Comment 13: “Conclusions: This section should report the main findings. The authors generally reported that they identified symptoms, but the reported information does not inform the reader about the main findings. I also suggest adding a paragraph with the study’s implications.”
Response 13: We agree. The Conclusion section now explicitly summarizes the main findings and adds a new paragraph outlining the clinical implications of routine PROM use and multidisciplinary management.
Comment 14: “References: Please revise references and provide the most recent when feasible. Several references are too old. The full stop should be placed after the references’ brackets. Please revise the entire text.”
Response 14: We agree. References have been carefully reviewed, with older citations updated where possible, and in-text citation formatting corrected to place the full stop after the reference brackets.
Response to Comments on the Quality of English Language:
Point 1: “The English is fine and does not require any improvement.”
Response 1: We thank the reviewer for this comment.
Additional clarifications: Line numbers might differ depending on Word version and definitions.
Reviewer 2 Report
Comments and Suggestions for Authors
in the abstract you write background but then write the aim... overall the abstract needs to be more readable.
Introduction is very small. Please provide more insight on the symptom burden on breast cancer. The importance of PROMS and then the ESAS. What is the gap that you need to fill? what is the novelty of the study.
2.1. Research design and procedures, setting and participants=make separate sections
Analyze ECOG and add reference.
Results are good but lessen the text since you have big tables.
Add footnotes in the tables eg BC=Breast cancer etc
At the discussion add comparison with systematic reviews on the matter.
Also you could state the importance of Multidisciplinary teams to asses and deal with symptoms.
Add also the strenghts of the study not just its limitations.
Add also a section on implications for clinical practice and future research.
Overall improve phrasing.
thank you
Comments on the Quality of English Language
Overall improve phrasing.
Author Response
Thank you very much for taking the time to review this manuscript. Please find the detailed responses below and the corresponding revisions in track changes in the re-submitted files.
Comment 1: “In the abstract you write background but then write the aim... overall the abstract needs to be more readable.”
Response 1: Agree. The abstract (lines 41 to 72) has been revised for clarity and readability. It now begins with a concise background statement, followed by the study aim, methods, results, and conclusions. The conclusions were also strengthened to emphasize the novelty and clinical implications of our findings.
Comment 2: “Introduction is very small. Please provide more insight on the symptom burden on breast cancer. The importance of PROMS and then the ESAS. What is the gap that you need to fill? what is the novelty of the study.”
Response 2: Agree. The Introduction has been substantially revised (lines 77 to 132). It now provides a more detailed discussion of the symptom burden associated with breast cancer, the importance of patient-reported outcome measures (PROMs), and the role of the Edmonton Symptom Assessment System (ESAS). We have also clarified the gap in the literature by noting that the only previous study relied on mean ESAS scores, which may underestimate clinically significant symptoms. Our study addresses this gap by applying established clinical thresholds in combination with patient-reported performance status. In addition, we improved phrasing throughout the Introduction to enhance clarity and academic tone.
Comment 3: “2.1.Research design and procedures, setting and participants=make separate sections.”
Response 3: Agree. We have reorganized the Methods section into separate subsections, with Research Design and Procedures (Section 2.1) and Setting and Participants (Section 2.2), to improve clarity and readability.
Comment 4: “Analyze ECOG and add reference.”
Response 4: Agree. We have revised Section 2.3 (Symptom and functional status assessment) to provide a clearer description of the Patient-Reported Functional Status tool and its relation to ECOG performance status. The revised text now reads (lines 177 to 182): “Functional status was assessed using the Patient-Reported Functional Status tool, a validated patient-completed version of the Eastern Cooperative Oncology Group (ECOG) performance status instrument. [16] This tool is a single-item measure with which patients rate their activity over the past month on a five-point scale ranging from 0 (“normal with no limitation”) to 4 (“pretty much bedridden, rarely out of bed”). The ECOG performance status is widely used by oncologists to quantify functional capacity and to estimate prognosis [36].”
Comment 5: “Results are good but lessen the text since you have big tables.”
Response 5: Agree. We have revised the Results section focusing on the most important findings while referring readers to the tables and figures for detailed information. This has reduced redundancy and improved readability.
Comment 6: “Add footnotes in the tables eg BC=Breast cancer etc.”
Response 6: Agree. Footnotes have been added to all tables to clarify abbreviations.
Comment 7: “At the discussion add comparison with systematic reviews on the matter.”
Response 7: Agree. We have revised the Discussion to include comparisons with findings from recent systematic reviews on symptom burden in breast cancer survivors, highlighting that our results are consistent with prior syntheses of the literature (lines 351 to 354).
Comment 8: “Also you could state the importance of Multidisciplinary teams to asses and deal with symptoms.”
Response 8: We agree with the reviewer and have added a dedicated paragraph to the Discussion emphasizing the importance of multidisciplinary teams in addressing both physical and psychological symptoms. This section highlights the roles of oncologists, nurses, psychologists, social workers, physiotherapists, and palliative care specialists in comprehensive symptom management (Lines 374 to 378).
Comment 9: “Add also the strenghts of the study not just its limitations.”
Response 9: Agree. The Discussion now includes a balanced appraisal of both strengths and limitations. Strengths highlighted include the relatively large cohort from two cancer centres, use of validated PROMs, application of clinically significant thresholds, and the integration of ECOG performance status to capture functional impact (Lines 379 to 384).
Comment 10: “Add also a section on implications for clinical practice and future research.”
Response 10: Agree. We have added a section outlining the clinical implications of our findings (lines 457 to 462), particularly regarding the routine use of PROMs with clinically significant thresholds and the integration of multidisciplinary care. We have also included directions for future research, including longitudinal symptom trajectories, ethno-racial differences, and the impact of PROM-driven interventions on outcomes (lines 463 to 467).
Comment 11: “Overall improve phrasing.”
Response 11: We thank the reviewer for this important observation. We have carefully revised the entire manuscript to improve phrasing, clarity, and consistency, ensuring a more concise, formal, and readable presentation of our study.
Response to Comments on the Quality of English Language:
Point 1: “The English could be improved to more clearly express the research.”
Response 1: We thank the reviewer for this helpful suggestion. The manuscript has been thoroughly revised to improve the quality of English, ensuring that all sections of the manuscript are expressed with greater clarity and precision.
Additional clarifications: line numbers might differ depending on Word version and definitions.
Round 2
Reviewer 1 Report
Comments and Suggestions for Authors
The authors have addressed all my previous comments and extensively reviewed the manuscript.
Reviewer 2 Report
Comments and Suggestions for Authors
Thank you